# Hurricane María tripled stem breaks and doubled tree mortality relative to other major storms

María Uriarte [1], Jill Thompson[2] & Jess K. Zimmerman[3]

Tropical cyclones are expected to intensify under a warming climate, with uncertain effects on tropical forests. One key challenge to predicting how more intense storms will influence these ecosystems is to attribute impacts specifically to storm meteorology rather than differences in forest characteristics. Here we compare tree damage data collected in the same forest in Puerto Rico after Hurricanes Hugo (1989, category 3), Georges (1998, category 3), and María (2017, category 4). María killed twice as many trees as Hugo, and for all but two species, broke 2- to 12-fold more stems than the other two storms. Species with high density wood were resistant to uprooting, hurricane-induced mortality, and were protected from breakage during Hugo but not María. Tree inventories and a wind exposure model allow us to attribute these differences in impacts to storm meteorology. A better understanding of risk factors associated with tree species susceptibility to severe storms is key to predicting the future of forest ecosystems under climate warming.

[1] Department of Ecology Evolution and Environmental Biology, Columbia University, 1200 Amsterdam Avenue, New York, NY 10027, USA. [2] Centre for Ecology & Hydrology Bush Estate, Penicuik, Midlothian EH26 0QB, UK. [3] Department of Environmental Sciences, University of Puerto Rico, San Juan, Puerto Rico 00925, USA. Correspondence and requests for materials should be addressed to M.U. (email: mu2126@columbia.edu)

Cyclonic storms (hurricanes, cyclones, and typhoons) represent the dominant natural disturbance in coastal tropical forests across the Caribbean, the Indian sub-continent, Southeast Asia, Indo-Malaysia, and northern Australia[1]. Since these storms derive their energy from ocean heat, and sea surface temperatures have increased in most regions of tropical-cyclone formation during the past decades, maximum wind speeds are projected to rise and storms to intensify[2], with some of the most significant increases in the North Atlantic[3–5]. Climate warming has also led to higher atmosphere moisture content, which is expected to increase tropical-cyclone rainfall rates[6]. Models predict that by 2100 in the North Atlantic basin, maximum sustained hurricane wind speeds will increase by 6–15%, coupled with increases of 20% in precipitation within 100 km of the storm center[7]. While it is difficult to attribute any specific storm to the effects of a warming climate[4,8], the extremely active 2017 hurricane season in the North Atlantic, with Harvey in Texas and Irma and María in the Caribbean and Florida, portends some of the projected shifts in hurricane regimes under a warming climate.

The expected changes in hurricane winds and rainfall may have profound consequences for the long-term resilience of tropical forests in the North Atlantic basin. The challenge to predicting how particularly severe storms, such as Hurricane María, will influence tropical forest ecosystems is to determine whether any change in observed tree damage and mortality could be attributed to differences in storm characteristics (i.e., wind speed and rainfall) rather than to differences in topographic exposure to wind or the structure and composition of forests at the time the storm struck. Here we employ tree damage and mortality data collected after three storms in a secondary tropical forest in Puerto Rico that developed after human disturbance during the first half of the 20th century[9]. We use these data to evaluate the effects of differences in wind speeds and tropical-cyclone rainfall rates on the forest and to identify the risk factors that moderated species vulnerabilities to storms of varying severities. Data derive from the 16-ha Luquillo Forest Dynamics Plot (LFDP) after three hurricanes: Hugo in 1989, Georges in 1998, and María in 2017 (Fig. 1a). At the time of landfall on the island, Hurricane Hugo was a category 3 storm with wind speeds of 166 Km/hr and total rainfall of ca. 200 mm[10]. Hurricane Georges was also a category 3 storm with wind speeds (144 Km/hr) and total rainfall (ca. 200 mm)[11] similar to those observed for Hugo. In contrast, Hurricane María struck the island as a category 4 storm with sustained winds up to 250 Km/hr and 500 mm of precipitation fell over 24 h[12]. María transformed tropical forests across the island into leafless tangles of damaged and downed trees (Fig. 1b). María was the strongest hurricane to make direct landfall in Puerto Rico since Hurricane San Felipe in 1928[13] and presages what climate warming will mean for hurricanes in the North Atlantic.

Our results show that María killed twice as many trees as Hugo, and for all but two species, broke 2- to 12-fold more stems than the other two storms. Extensive tree inventories and a wind exposure model allow us to ascribe these differences in impacts to variation in storm meteorology (i.e., wind speed and rainfall).

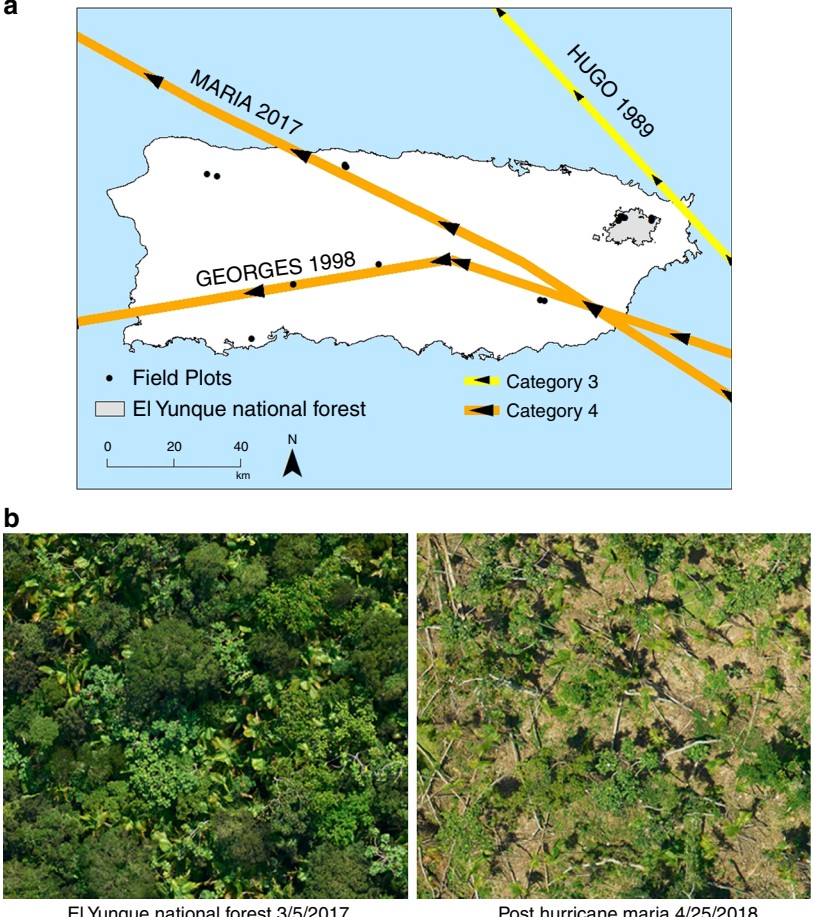

**Fig. 1** Tracks of Hurricanes Hugo, Georges, and María (**a**); and aerial photographs of the study area in El Yunque National Forest before and after Hurricane María (**b**). Images courtesy of D. Morton, NASA

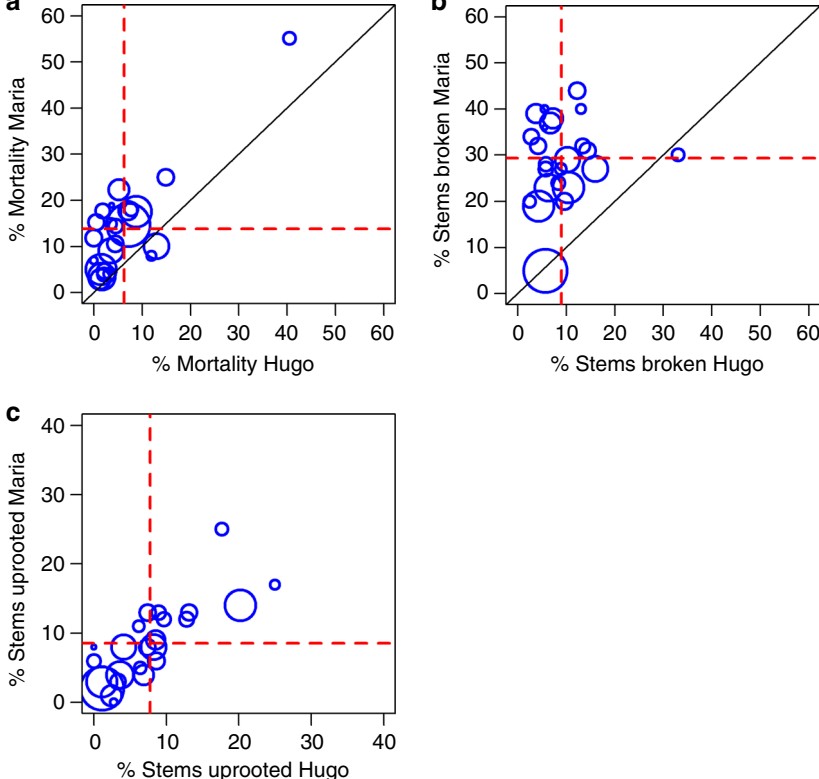

**Fig. 2** Rates of immediate mortality (**a**), stem break (**b**), and uprooting (**c**) as a result of Hurricanes Hugo (1989) and María (2017) for 24 tree species. Size of circles is proportional to the number of stems. Dashed red lines indicate community-average rates

Tree species with high density wood were particularly resistant to uprooting and hurricane-induced mortality in all storms, but were protected from breakage during Hugo but not María. Large trees were also more likely to break in María but not Hugo. Taken together, these results suggest that more severe storms expected under a changing climate can alter species composition and size composition of these forests.

### Results

**Stem damage and mortality.** Hurricane María killed twice as many stems as Hugo in the LFDP (Fig. 2, Supplementary Table 1–3). The proportion of uprooted stems, however, was similar for Hugo and María but lower for Georges. The most striking difference between María and the other two hurricanes was a 2- to 12-fold increase (average = 3.27-fold) in species-specific stem break rates for all but two species, the palm *Prestoea acuminata var montana*, and the pioneer species *Cecropia schreberiana* (Supplementary Table 3). Despite differences in history, structure and composition of the forest was remarkably similar ahead of each of these hurricanes. The large disparity in the impacts of María compared to the other two storms cannot be explained by differences in tree diameter sizes in the forest at the time of impact (Supplementary Figures 1 and 2, Supplementary Table 4). Exposure of each tree in the LFDP to wind was calculated using the EXPOS model (see Methods) for the three hurricanes and exposure of the forest to storm winds was far greater during Hurricane Hugo than during Georges or María (Supplementary Figure 3) because of the track of the storms across the island and the position of the study site (Fig. 1a). Consequently, plot position relative to the storm track does not account for the observed differences in damage among storms.

Models of the probability of individual stem breakage or mortality that incorporated diameter size and wind exposure further reinforce the notion that differences in the effects of the three storms did not reflect differences in forest structure at the time of impact nor in topographic exposure to hurricane-force winds (Supplementary Table 5). Rather, differences in rates of stem break reflect to a large degree higher vulnerability of large diameter trees during H. María, suggesting that differences in the meteorological characteristics of the hurricanes were responsible for the striking differences in rates of stem break among storms (Fig. 3, Supplementary Table 6).

**Selective pressure of hurricanes on tree species.** Hurricanes exert selective pressure on forests by damaging some species more than others[14]. Insight into the drivers of the observed disparities in damage between the three storms can be derived from examining variation in species responses to the storms. At the study site, the only two species to exhibit similar responses to the three hurricanes were the palm *P. acuminata* and the pioneer species *C. schreberiana*; the palm had the lowest rates of stem break in both storms and *C. schreberiana* had the highest (Supplementary Table 3). Rates of damage for most species were generally lower for H. Georges (Supplementary Table 3) but this hurricane damaged a smaller area of the forest than Hugo and María and we had a less damage data for Georges. The abundance of *P. acuminata* doubled, and *C. schreberiana* quadrupled between 1995 and 2016 (Supplementary Table 1), suggesting that Hurricanes Hugo and Georges increased the number of individuals of these species in the forest.

We examined the association between species-specific characteristics, namely maximum tree height, specific leaf area, and wood density, and the rates of mortality and modes of

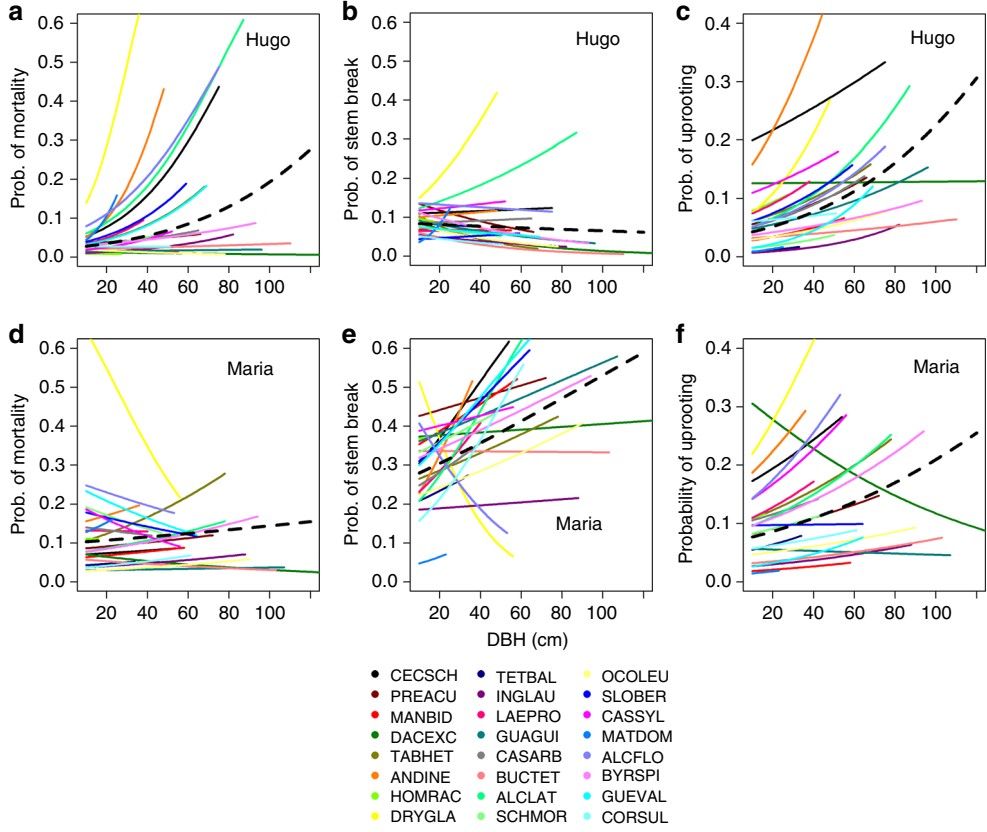

**Fig. 3** Relationship between species diameter and rates of stem break (**a**, **d**), uprooting (**b**, **e**), and mortality (**c**, **f**) in H. Hugo and María. Black dashed lines depict community-wide rates. Individual species estimates are restricted to the maximum observed dbh in the data

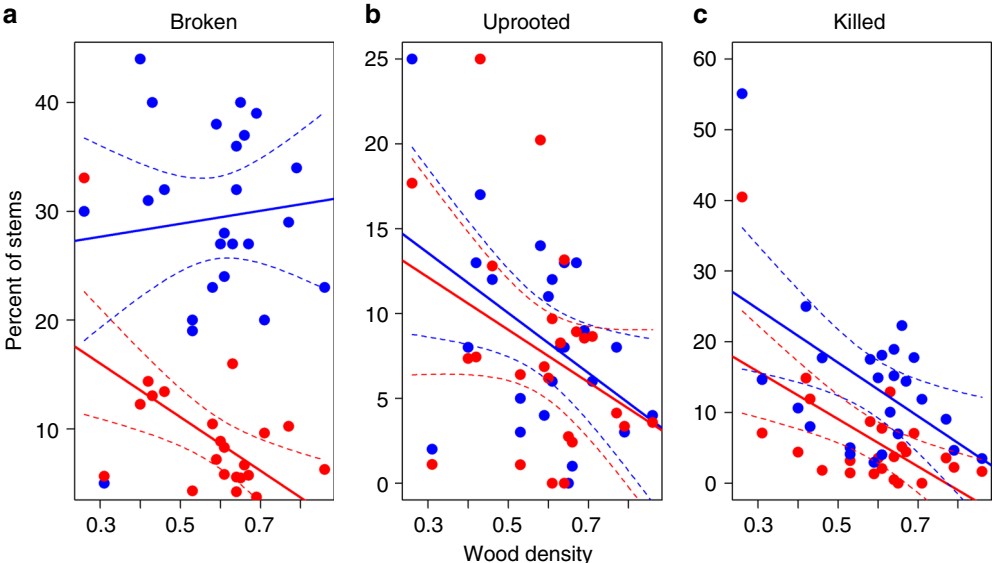

**Fig. 4** Relationship between wood density and rates of stem break (**a**), uprooting (**b**), and immediate mortality (**c**) for 24 tree species during Hurricanes Hugo (red dots) and María (blue dots). Dashed lines indicate 95% confidence intervals

damage in Hurricanes Hugo, Georges, and María. Species with high wood density suffered lower immediate mortality during both Hugo and María and were less likely to uproot during María, or break during Hugo (Fig. 4, Supplementary Table 7). During H. María, however, dense wood did not afford species any protection from stem break as there was no differences between rates of stem break between species with high and low-density wood. Rates of stem break, uprooting, and mortality over periods free of storms were also unrelated to wood density (Supp. Tables 3 and 7).

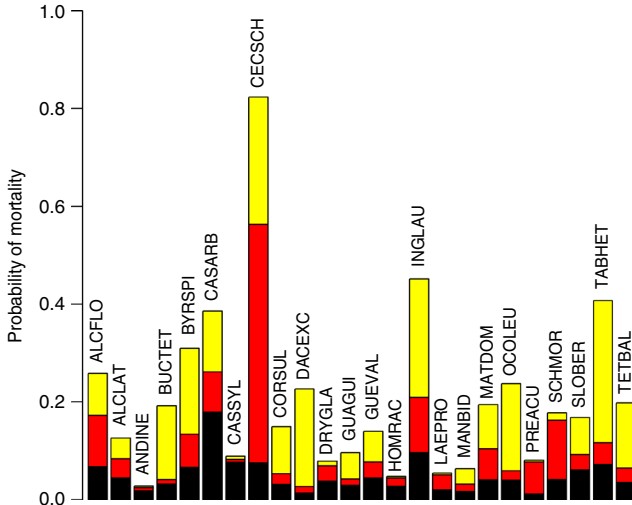

**Fig. 5** Background (black) and hurricane-induced mortality, immediate (red) and delayed (yellow), from Hugo. Delayed mortality rates are estimated for trees ≥ 10 dbh that were present at the time of the storm and damaged and died between the completion of the damage assessment in 1991 and the 1995 census. Background mortality represent tree mortality over the same period for undamaged trees

**Lagged tree mortality from hurricane damage**. Windstorms are prominent causes of tree mortality[15]. Background mortality rates for trees > 10 cm diameter at breast height in wet tropical forests range between 1 and 2% per year and rarely exceed 3%[15–17]. Because coastal and island tropical forests often experience large catastrophic windstorms, species are adapted to these disturbances and tree mortality rates even after a severe windstorm tend to be low, ranging between 7 and 12%, only 2–3 times background mortality rates[15,18]. Wind induced mortality, however, is often delayed and results from branch and canopy damage during storms[19–21]. To estimate potential rates of delayed mortality resulting from stem breakage during María, we used census data collected in 1995 after the passage of Hurricane Hugo. In the 5 years following Hurricane Hugo, 30% of broken stems and 55% of those uprooted during the storm died (Fig. 5 and Supplementary Table 8). Although these numbers to some degree reflect stand dynamics following the storm[20], they provide a reasonable estimation of the likely effects of damage from María on the fate of severely damaged trees.

## Discussion

The large disparity in the impacts of María compared to the other two storms cannot be explained by differences in tree diameter sizes in the forest at the time of impact or differences in wind exposure. Hurricane María combined extreme precipitation and strong winds that may have reduced soil stability and cohesion and thus root anchorage, while simultaneously exerting a strong dynamical force on the stem and crown. Dynamic wind loadings typically drive roots to bend and twist as a result of the rotational pivoting of the trunk, and if the wind is strong enough, it will lead to uprooting[22]. Pre-hurricane soil moisture in particular is a major controlling factor in the nature of damage (uprooting vs. stem breakage)[23]. In dry soils, stem breakage is the dominant type of damage while in wet soils uprooting is more common[23]. Given the fact that H. Hugo and Georges were considerable drier than María, it is surprising that uprooting rates were not higher during María than the other two storms. This observation coupled the sharp increase in stem breaks during H. María, particularly for larger trees, may have been driven by the passage of Hurricane

Irma, a category 4 storm that skirted the north of the island on 7 September 2017, 2 weeks before 20 September, when María struck the island. Although Irma did not make landfall on Puerto Rico (Fig. 1), it removed a substantial amount of tree foliage (Zimmerman, pers. obs), possibly reducing wind drag forces over the canopy and how such forces are transmitted to the base of the tree, favoring stem break over uprooting[22].

The impact of each of the severe hurricanes that have affected the Luquillo forest between 1989 and 2017 may not be independent. The amount of damage depends on the state of the forest after the previous hurricane and the degree of recovery from prior storms. In particular, trees present when Hugo struck the forest had developed over 57 years without hurricane damage and likely had larger crowns with greater wind resistance that results in greater damage (e.g., uprooting). Although diameter distributions for the majority of species did not differ significantly between H. Hugo and Georges, it is likely that some of the old and large trees damaged during H. Hugo had smaller crowns and were then less affected by H. Georges, possibly accounting for lower rates of damage during this storm. Twenty years had passed between H. Georges and María, allowing the canopy to recover. Nevertheless, loss of foliage to H. Irma may have shaped the type of damage (i.e., a greater number of stem breaks) María inflicted on the forest. It is also plausible that other factors (e.g., biomechanical properties of tree species) underlie the different impacts observed across the three storms.

Species vulnerability to hurricanes depends on the strength of wood, stem diameter, shape and size of the crown, leaf features, and extent and depth of root system[24–26]. Species with weaker wood[27], higher canopy-diameter ratios, and shallower root systems[28] generally suffer greater damage and mortality. Previous work after Hurricane Hugo and Georges at the site examined the relationship between damage and successional specialization and our results corroborate those findings: pioneer species with low-density wood are generally more vulnerable to hurricanes than old-growth high wood density specialists[24,25]. Our results however, also demonstrate that species with high density wood and large trees, although largely resistant to breakage in less severe storms, may be vulnerable to stem break during hurricanes of María's severity. Given high rates of mortality of broken and uprooted stems, it is likely that trajectories of forest recovery from H. María will substantially diverge from recovery from Hugo, with consequences for both size structure and species composition of the forest.

Predicting the effects of climate warming on wind disturbance regimes and their effect on forest faces a number of challenges[29]. The susceptibility of forest ecosystems to wind damage is determined by tree and stand characteristics as well as site factors[23]. A number of studies have identified factors for tree species[24–26] but our understanding of the biomechanical characteristics of trees and how they relate to other aspects of tree life histories and physiology, is extremely limited. Successional specialization is of particular interest because second-growth forests account for over 70% of forest cover in tropical regions[30] and generally have a greater proportion of low wood density pioneer type species that may be more prone to storm damage. Predicting how these ecosystems will fare under a warming climate will require a more nuanced understanding of the relationship between successional specialization and the biomechanical characteristics that moderate species and forest vulnerability to hurricanes. A second, and perhaps more urgent issue that we highlight in this paper is that risk factors associated with resistance to an 'average' storm (i.e., high density wood) may be quite different than those mediating impacts of the more severe storms forecasted under warming. Cyclonic storms select for windstorm resistance [e.g.[14],] and the rapidity of change in storm disturbance regimes may exceed the

capacity for adaptation of the forest communities[31], yielding a depauperate forest dominated by a few pioneers (e.g., *C. schreberiana*) and wind resistant species (e.g., palms) that can withstand high severity storms. Our results also demonstrate that large trees are particularly vulnerable to storms of María's severity, presaging potential shifts in the size structure and carbon storage potential of these forests.

## Methods

**Study site**. The Luquillo Forest Dynamics Plot (LFDP) is a 16-ha permanent, mapped forest plot located in the Luquillo Mountains of northeastern Puerto Rico (SW corner 18° 20' N, 65° 49' W). The forest is classified as subtropical wet in the Holdridge life zone system[32]. Vegetation and topography of this research area are typical of the tabonuco (*Dacryodes excelsa*) forest zone. Rainfall averages 3500 mm per year. Elevation ranges from 333 to 428 m a.s.l. All of the soils are formed from volcaniclastic rock. The forest has experienced substantial natural and human disturbances during the past century. Prior to 1934, parts of the LFDP were subjected to light logging and agriculture, but the forest structure and canopy cover had substantially recovered when in 1989, after a period of 57 years with no major storm, Hurricane Hugo struck the forest[9]. Basal area was estimated to average 36.7 m$^2$ ha$^{-1}$ at the time of hurricane Hugo, 30.85 m$^2$ ha$^{-1}$ at the time Georges struck, and 38.37 m$^2$ ha$^{-1}$ in 2016, the year before María struck the forest.

**Tree data**. Tree damage data in the LFDP was collected over a maximum 15 months following each of the three hurricanes. The surveys recorded several qualitative and quantitative observations on tree damage resulting from the hurricane, such as uprooting or stem break, and any type of damage to stems, tree crowns and branches (see ref. [24] for details). To assess damage and immediate mortality from H. Hugo, a survey of all trees ≥ 10 cm diameter at breast height (dbh = diameter at 1.3 m from the ground) was conducted between 1990 and 1991. All trees in the plot were surveyed again in 1995–1996 and we used these data to assess the effects of damage during Hugo on delayed mortality. Following Hurricane Georges in 1998 damage to woody stems ≥ 10 cm dbh was assessed in 40 subplots (20 × 20 m in size) in a grid pattern (60 m spacing) distributed regularly across the LFDP rather than for the entire plot. In 2018, we surveyed damage and immediate mortality from Hurricane María for all trees ≥ 10 cm dbh in the LFDP.

Wood density (g$^{-1}$cm$^{-3}$) and specific leaf area (SLA, cm$^2$g$^{-1}$) measurements were collected for at least 10 individuals per species using standard protocols[33]. Briefly, leaf area was measured on sun-lit foliage of mature individuals and leaves were dried for 48 h and weighed to calculate specific leaf area (SLA). For wood density calculations tree cores were extracted, measured for volume, and oven-dried before weighing. For all analyses, we used the mean value for each trait for each species.

**Analyses of hurricane tree damage and immediate mortality**. To compare the effects of the three hurricanes on tree species damage and mortality, we selected species with at least 40 stems in the damage assessment. This criterion yielded 24 tree species for Hugo (88% of stems assessed), María (81%) and 12 for Georges (89%) (Supp. Table 1). We then compared the percentage of stems for each species that were uprooted, broken or, for Hugo and María, likely killed by the hurricanes. To eliminate the possibility that differences in damage in the three hurricanes were driven by variation in stem diameter size distributions or wind exposure, we conducted two sets of analyses. We first compared size diameter distributions and topographic wind exposure for each species for the three hurricanes using a χ$^2$ test. Exposure to hurricane winds for each tree during each storm was estimated using a topographic model (EXPOS), which determines the degree of exposure to winds given hurricane track and wind speed data, using a 5 m resolution, LiDAR-derived digital elevation map (DEM)[34]. The model assumes that hurricane movement over land decreases sustained wind speeds and increases inflow angles, and then calculates spatial variation in exposure at the spatial resolution of the DEM. This model has been shown to accurately reconstruct exposure to hurricane winds in Puerto Rico at the landscape scale, when compared to historical records[13,34]. Track and wind speed data for the three hurricanes was obtained from NOAA and the LiDAR-derived 5 m Digital Elevation Model was obtained from the USGS 3DEP elevation program (https://nationalmap.gov/3DEP/). In a second analysis, we fitted mixed models of the probability of severe damage (stem break and uprooting) and immediate mortality for individual stems that incorporated tree diameter and wind exposure information with species as random effects. Random effects for this model estimate species variation in average rates of hurricane mortality or damage for each storm after accounting for the effects of variation in stem diameter size or exposure. As a result, these random effects allow us to estimate variation in species damage or hurricane mortality that can be attributed to differences in the meteorological characteristics of the storms. To prevent confounding of species-specific damage and mortality with inter-specific variation in tree diameter size, we standardized diameter size within species for each storm by subtracting species-specific means from individual stem values. Models were fitted using the "lme4" package in R Statistical software[35].

**Analyses of lagged, hurricane-induced tree mortality**. Beyond its effects on immediate mortality, biomass loss and damage during a hurricane also leads to delayed mortality[19]. These effects typically play out over 3–5 years[1,19]. To evaluate the effects of the hurricanes on delayed tree mortality, we used data collected in the 1995–1996 census and mixed models to estimate the increase in probability of delayed mortality that could be attributed to stem breakage or uprooting damage, which was above background mortality rates. Random effects for species were included as intercepts (background rates) and as slopes for the damage effect on mortality. Background rates estimate probability of mortality between 1990 and 1995 that could not be attributed to damage suffered in H. Hugo. Species-specific random slopes for effects of severe damage (stem break or uprooting) on delayed mortality recorded in 1995 estimate increases in probability of mortality that can be attributed to severe damage (i.e., stem break or uprooting) during H. Hugo. Severe damage was coded as a binary variable (0, 1). Note that background rates over this period are likely overestimates relative to storm-free periods since even if not broken or uprooted, stems suffered crown damage that could affect subsequent survival[20].

**Reporting summary**. Further information on experimental design is available in the Nature Research Reporting Summary linked to this article.

## Code availability

Code used for analyses is available upon request.

## Data availability

All damage and census data collected between 1990 and 2016 is available in the Luquillo LTER data catalog http://luq.lter.network/datacatalog. Damage data collected after Hurricane María is available upon request with some restrictions.

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

## Acknowledgements

We thank the field crews who collected tree census and damage data. Research was supported by NSF award DEB-1801315 to M.U. and J.K.Z. and awards DEB-9705814, DEB-0080538, DEB-0218039, BSR-8811902, DEB-9411973, DEB-0620910, DEB-1239764, and DEB-1546686 to the Luquillo LTER. Gabriel Arellano was instrumental in collecting post-María Hurricane data, as part of the Next Generation Ecosystem Experiments-Tropics, funded by the U.S. Department of Energy, Office of Science, Office of Biological and Environmental Research.

## Author contributions

M.U. conceived the paper, analyzed the data, and wrote the first draft. J.K.Z., J.T., and M.U. collected census and damage data and contributed to writing.

## Additional information

**Competing interests:** The authors declare no competing interests.

