## [Peer Review File · Nature Communications]

Reviewers' Comments:

Reviewer #1:

Remarks to the Author:

This manuscript reports immediate damage and mortality caused by a very intense tropical cyclone to a long-term research site in a tropical forest in Puerto Rico and compares these immediate effects with that of a less intense tropical cyclone that was documented similarly, 28 years earlier, in the same forest. The study's essential message that damage and mortality is greater in a more intense cyclone is unsurprising.

I think the manuscript would be better if it used all available data to determine how the damage and mortality caused to trees by tropical cyclones scales is related to cyclone intensity. As the authors had previously noted (Canham et al. 2010, cited), "one of the most significant challenges in developing a predictive understanding of the long-term effects of hurricanes on tropical forests is the development of quantitative models of the relationships between variation in storm intensity and the resulting severity of tree damage and mortality". Given those challenges, I think this manuscript would benefit by bringing to bear all data from the study site across a tropical cyclone intensity gradient and the context of the disturbance regime that affects the well-studied tropical forest reported in this manuscript. Notably, it is surprising that the effects of Hurricane Georges (category 3) of 1998, which occurred after Hurricane Hugo, were not included in this manuscript. In an earlier paper by this manuscript's authors (Urrate et al. 2005), they noted that "the forest canopy in the Luquillo forest was returning to pre-Hugo height and evenness when it was severely damaged by Hurricane Georges in September 1998 (Brokaw et al. 2004)" (Uriarte et al. 2005). Beard et al. (2005) note the (mostly minor) influences of several other tropical cyclones since Hurricane Hugo. With only two points of comparison (an intense and a very intense cyclone), I think the manuscript is not all that it could be. A reader does not learn whether effects scale linearly or otherwise with increasing disturbance intensity (e.g., did Hurricane Hugo also "triple stem breaks and double tree mortality" compared with Hurricane Georges?).

If this were better developed, it would be interesting to know, in terms of the wider generality of the study, whether the tree functional traits that were predictors of damage and mortality in the cases of two cyclones of different intensity (p4, lines 17–21) were useful across a wider intensity gradient, whether they scaled linearly with cyclone intensity or whether there was a threshold of cyclone intensity above which species with low wood density are more apt to be killed or uprooted. Clearly each cyclone that affects a forest is non-independent; effects are likely to be interactive, contingent on some combination of intensity of each cyclone, the interval between them, and, as the authors note, rainfall and duration, so these caveats will be needed.

Another important point that is surprisingly omitted in the interpretation of two cyclones on the Luquillo 16-ha plot is the imprint of past human disturbance. Comita et al. (2010) note that about two-thirds of the plot was "subjected to high intensity historical human land use and, as a result, is dominated by trees of secondary-forest species". Clearly therefore there is likely to be an interaction between cyclone disturbance and past human land use (e.g., forests comprised mostly of "secondary-forest species" are likely to be biased towards tree species with low wood density). This caveat is needed in the current manuscript and in discussions of the generality of findings and implications for "new forests" (p6, line 14).

There is a consensus that intense tropical cyclones will likely become more intense because of anthropogenic climate change, and there is some evidence that large cyclones may move more slowly with greater rainfall attending them (e.g., van Oldenborgh et al. 2017). Nonetheless, I felt that the authors used terms like "portends" (p2, line 18) and "presages" (p3, line 13) rather incautiously,

given the strength of some past Caribbean hurricanes. The authors note the San Felipe hurricane of 1928 (p3, line 13), and Crow (1980) noted that the composition of forests of the Luquillo Mountains was likely to be shaped not only by that hurricane but also by the San Nicolas and San Ciprian hurricanes of 1931 and 1932. Seen in these terms the effects on intense hurricanes likely happened in the past (see also Figure 1 of Boose et al., 2004, cited cf. Figure 1a of this manuscript). The “extremely active 2017 hurricane season” in the Caribbean is hardly unprecedented either: the likely most intense Caribbean cyclone of the last four centuries (Hurricane San Calixto of 1780) occurred in an active season for hurricanes (all in October that year). It is moot whether “new forests” (per line 6, line 14), affected by a series of intense cyclones, had already developed in response to the cyclone disturbance regime of the 1780s.

Minor points:

Terminology: the text alternates between “hurricanes” (a regional term) and “tropical cyclone” (the general meteorological term), sometimes sentence by sentence (e.g., p2, lines 8 vs. 10).

It is untrue that “hurricanes represent the dominant natural disturbance in coastal forests across the Americas” (p2, line 8); cyclone disturbances are very rare in the south Atlantic and most coastal South American forests never experience cyclone disturbance. As for “coastal forests ... in other tropical regions” (p2, line 9), many are unaffected by cyclones (e.g., tropical West Africa).

I think the title would be more accurate if it sought to generalize less. Because Hurricane María was a single event, the title would be more apt as “A stronger, wetter hurricane tripled stem breaks and doubled tree mortality in a tropical forest” although, even here, the comparative adjectives are problematic. “Stronger and wetter” than what? (see earlier comments about the cyclone disturbance regime).

Re damage to palms, per earlier comments about using all available data, see Zimmerman & Covich (2007) with respect to Hurricane Georges. Also, re Figures 4, S1, S2, and S4, “PREACU” rather than “PREMON” to be consistent with the main text, and Table S1, *Prestoea acuminata*.

Tables S1 and S2: Species name is missing for the *Guettarda* species (GUEVAL) and for the *Tetragastris* species (TETBAL). For Table S1, www.theplantlist.org lists all of *Casearia*, *Homalium*, and *Laetia* as *Salicaceae*.

Table S2: I presumed the first three columns related to Hurricane Hugo and the next three to Hurricane María, but another row denoting this is needed. I think Table S2 would be improved if it listed values for these species’ wood densities (cf. p4, line 17) and root bole areas (cf. p4, line 21, for the 11 pertinent species).

P10, reference #15, should be Frangi JL, Lugo AE as authors.

P11, reference #23 should be Lugo AE, Scatena FN as authors.

P12, reference #32: something odd with the author listing.

References

- Comita LS et al. 2010 Interactive effects of land use history and natural disturbance on seedling dynamics in a subtropical forest. *Ecological Applications* 20, 1270–1284.
- Crow TR 1980 A rainforest chronicle: a 30-year record of change in structure and composition at El Verde, Puerto Rico. *Biotropica* 12, 42–55.
- Uriarte M et al. 2005 Seedling recruitment in a hurricane-driven tropical forest: light limitation, density-dependence and the spatial distribution of parent trees. *Journal of Ecology* 93, 291–304.
- van Oldenborgh GJ et al. 2017 Attribution of extreme rainfall from Hurricane Harvey, August 2017. *Environmental Research Letters* 12, 124009
- Zimmerman JKH, Covich AP 2007 Damage and recovery of riparian Sierra palms after Hurricane

Georges: influence of topography and biotic characteristics. *Biotropica* 39, 43–49.

Reviewer #2:

Remarks to the Author:

This paper presents a novel analysis comparing hurricane damage to trees within a large forest dynamics plot following hurricanes Hugo in 1989 and Maria in 2017. The analysis shows that stronger, wetter hurricanes such as Maria do a lot more damage and identifies some species' characteristics that affect vulnerability. Overall, this is an important and timely analysis. There are some relatively minor issues that need to be addressed:

To put the stand-level comparisons in context, it would be helpful to know more about how the pre-Hugo and pre-Maria stands compared in terms of successional stage/ composition/ history. When was the last major hurricane before Hugo? Was the stand at a similar successional stage for the two hurricanes, with similar size distributions for all species combined, maximum tree sizes, and basal area/stand-level biomass? The analyses in Fig. S2 provide convincing evidence that the size distributions of most of the focal species were similar, and it would be nice to have more context behind the stand-level statistics (e.g., paragraph starting p. 3).

Please be more clear in the presentation of mortality rate. It's often confusing whether numbers presented are immediate or total (including delayed) mortality. Furthermore, unless I'm missing something, the paper doesn't actually present the total percent mortality. Please present % immediate and delayed mortality for both storms. Similarly, please present total % stem breakage and uprooting.

This is not a requirement for the paper to be acceptable for publication and may be subject for a separate publication, but it would be interesting to know how damage and mortality varied with tree size (DBH, or height if possible).

Regarding the analysis presented in Figure 4... Presumably, root bole area scales with tree size, so a species mean is likely highly dependent upon the size distribution of trees sampled. Does this match the size distribution within your analysis? Furthermore, to what extent is this relationship driven by root bole area, as opposed to DBH/ height? Would you see a similar relationship if you normalize by DBH (i.e., ratio of root bole area to basal area)?

Minor comments:

-wherever you present results related to mortality (e.g., Fig2a, 3c), specify whether its current or projected total mortality.

p.2, line 17- you could add 2018..

-Fig. S3- why is wind exposure so different? (Please explain what determines the topographic exposure index.)

-p. 8, line 18- from where are background rate estimates obtained?

-p. 8, line 16-23- Please be a little more specific/ clear as to how lagged mortality was estimated (e.g., what were the terms in the model, how were model estimates applied to estimate post-Maria mortality).

Reviewer #3:

Remarks to the Author:

This manuscript describes an interesting dataset, comparing damage in the same forest area after two hurricanes with differences in intensity: Hugo, 1989 and Maria 2017. The premise of the work is that understanding the differences in damage between storms with different wind speeds and rainfall will help us understand how forests will be affected by intensified cyclones that are expected with climate change. I have a number of issues with the way that the data has been collected and is analysed and presented in this manuscript.

General issues

1. Firstly, the title implies that stronger, wetter hurricanes will always triple stem breaks and double tree mortality, while all we have to go on is the difference between two hurricanes. A more appropriate title might be "A comparison of tree breakage and mortality following two hurricanes".
2. The paper uses terminology that is not always clear:
When you say that you assessed mortality in terms of trees uprooted, broken or immediately killed, how did you assess that trees had been killed? Trees can be very badly damaged and still resprout the following year. The second survey of mortality six years after Hugo could clarify how accurate the initial assessment was, if the same individual trees were identified in both, but there is nothing to indicate that this was done.
3. The term 'tipped up' used on p3 line 17, p7 line 7 and 10, p15 Figure 2 caption is not clear. Do you mean overturned (both broken and uprooted), or just 'uprooted'? If it means 'uprooted' please just use that term consistently.
4. How do you define the 'root bole'?
5. It is not clear what has been measured: In the Methods you say that the trees surveyed following Hugo had stem dbh > 10 inches, while after the Maria storm surveyed trees were > 10 cm. This is a big difference - Is this an error in the text or an error of the methodology?
6. p7, line 10 states that 'we also measured bole size', but doesn't give any details of how this was done. These are complicated structures and would presumably need a consistent measurement system to provide dimensions such as 'bole' spread, root spread, surface area, soil maximum and mean depth, volume etc. What was measured and how?
7. Please include more details of the inputs needed for the topographic model (p7 line 22), and the scale at which it was applied? Was 'exposure' calculated for each individual tree? Does the exposure value represent exposure through the life of the tree as well as to the damaging hurricane?
8. In the Main Text there is a statement that max tree height, leaf area, wood density and root bole area were used in a comparison with mortality / mode of damage. However, the Methods section does not describe how these characteristics were measured. Were they measured on a sample of trees? What sample size? How were they selected? How were they measured? Was it green density you measured?
9. The analysis method does not always appear appropriate. The paper describes a relationship between the change from Hugo to Maria in proportion of stems broken and the "Species mean root bole area" (Figure 4). However, species bole area means were calculated from trees uprooted (in just one event: Maria), while the broken trees have no measured root bole area. So you are assuming that the root bole dimensions of uprooted trees tells you something about the root system size of broken trees. As trees break when their breakage moment is reached before their uprooting moment, root or root bole characteristics may be expected to differ between broken and uprooted trees. So the broken

trees could have much wider stronger root systems, or perhaps much narrower deeper ones than the uprooted trees. This will vary between species and soils. Therefore the comparison in Figure 4 does not appear to have validity. To understand the increase in stem breakage in terms of root or root bole characteristics, you would need to have root characteristics from both broken and uprooted trees after both events. There may however be some explanation of the difference in terms of the different distribution of tree sizes between species and events. The BUCCAP and DACEXC species that showed the largest change in proportion of broken stems (Fig 4) appear to have a larger mean increase of stem diameter between events than other species (Fig S2), but unfortunately those differences were not significant.

10. The paragraph (p5) on possible explanations for differences in failure mode appears confused as to whether you expect wet soil to improve or decrease anchorage. Some authors indicate that very wet soils have reduced root-soil cohesion and therefore tree anchorage should be reduced. As the soil under a root plate can shear from the soil under the plate, before roots fail, and the soil weight is just one component of anchorage, it seems unlikely that the increased weight of a wet root plate will make a big enough difference to anchorage to change the failure mode from uprooting to breakage. Maybe it could, but it would be worth developing an argument by referring to the tree anchorage component work by Coutts from the 1980s. You assume that there really was a difference in soil water content during these two events, but no evidence is presented on this.

11. The analysis could be linked with existing work on tree stability and risk. For example, there is no attempt to use or discuss the relevance of published models of root-soil plate/bole dimensions and tree anchorage, or models of wind risk in relation to tree dimensions. Root-bole soil weight increase with higher rainfall could be examined in one of these models to examine the effect on turning moment, to explore your proposal (p5 para 1) that saturated soil in boles led to increased stem breakage. Wind risk models might allow prediction of the increased damage from the observed difference in wind speeds between the hurricanes, and might help understand differences in damage in terms of tree height and other measured characteristics.

Specific points

12. p1 Abstract. References are not usually included in an Abstract. Please remove these.
13. p4 line 2. Change "in exposure to hurricane-force winds" to "in topographic exposure" ?
14. p4. line 5. Do you mean 'favouring some individuals' rather than 'some species'?
15. p4, line 11. strength of the wood
16. p6, line 14-16. Where is the evidence that the reduced species mix would provide reduced C sequestration, wildlife habitat, and other ecosystem services?

Reviewers' comments

Reviewer #1 (Remarks to the Author):

This manuscript reports immediate damage and mortality caused by a very intense tropical cyclone to a long-term research site in a tropical forest in Puerto Rico and compares these immediate effects with that of a less intense tropical cyclone that was documented similarly, 28 years earlier, in the same forest. The study's essential message that damage and mortality is greater in a more intense cyclone is unsurprising.

I think the manuscript would be better if it used all available data to determine how the damage and mortality caused to trees by tropical cyclones scales is related to cyclone intensity. As the authors had previously noted (Canham et al. 2010, cited), "one of the most significant challenges in developing a predictive understanding of the long-term effects of hurricanes on tropical forests is the development of quantitative models of the relationships between variation in storm intensity and the resulting severity of tree damage and mortality". Given those challenges, I think this manuscript would benefit by bringing to bear all data from the study site across a tropical cyclone intensity gradient and the context of the disturbance regime that affects the well-studied tropical forest reported in this manuscript. Notably, it is surprising that the effects of Hurricane Georges (category 3) of 1998, which occurred after Hurricane Hugo, were not included in this manuscript.

In an earlier paper by this manuscript's authors (Uriarte et al. 2005), they noted that "the forest canopy in the Luquillo forest was returning to pre-Hugo height and evenness when it was severely damaged by Hurricane Georges in September 1998 (Brokaw et al. 2004)" (Uriarte et al. 2005). Beard et al. (2005) note the (mostly minor) influences of several other tropical cyclones since Hurricane Hugo. With only two points of comparison (an intense and a very intense cyclone), I think the manuscript is not all that it could be. A reader does not learn whether effects scale linearly or otherwise with increasing disturbance intensity (e.g., did Hurricane Hugo also "triple stem breaks and double tree mortality" compared with Hurricane Georges?).

Response: We have included data recorded after the passage of Hurricane Georges in 1998. However, damage data from this storm was only collected for a subset (ca. 20% of stems in the 16-ha plot) so the comparisons to other storms can only be conducted for those species that had at least 40 stems assessed for damage. For species with fewer stems, we cannot conduct robust statistical analyses. The new analyses do not change the storyline but they do provide context to interpret the impacts of María. As the reviewer points out, the site has also been subject to a number of smaller storms. To account for the potential impacts of these smaller storms on tree damage and mortality, we also provide annualized tree break, uprooting, and mortality rates during a 5-year period (2011-2016) that was free of severe storms.

If this were better developed, it would be interesting to know, in terms of the wider generality of the study, whether the tree functional traits that were predictors of damage and mortality in the

cases of two cyclones of different intensity (p4, lines 17–21) were useful across a wider intensity gradient, whether they scaled linearly with cyclone intensity or whether there was a threshold of cyclone intensity above which species with low wood density are more apt to be killed or uprooted. Clearly each cyclone that affects a forest is non-independent; effects are likely to be interactive, contingent on some combination of intensity of each cyclone, the interval between them, and, as the authors note, rainfall and duration, so these caveats will be needed.

Response: We now report the relationship between species traits and the different damage types in the three storms and also under background conditions. Rates of stem break were only associated with wood density for Hugo but not the other two storms. Given the similarity in the meteorology of Hugo and Georges, such association may also have been present in Georges but we do not have the data to conduct a sound analysis for this second storm. For the species with sufficient stems assessed for damage, stem break rates in Georges were far lower than in Hugo. Rates of stem break, uprooting, and mortality over quiescent periods (2011–2016) were not associated with wood density.

Another important point that is surprisingly omitted in the interpretation of two cyclones on the Luquillo 16-ha plot is the imprint of past human disturbance. Comita et al. (2010) note that about two-thirds of the plot was “subjected to high intensity historical human land use and, as a result, is dominated by trees of secondary-forest species”. Clearly therefore there is likely to be an interaction between cyclone disturbance and past human land use (e.g., forests comprised mostly of “secondary-forest species” are likely to be biased towards tree species with low wood density). This caveat is needed in the current manuscript and in discussions of the generality of findings and implications for “new forests” (p6, line 14).

Response: Previous work has examined the relationship between damage and successional specialization and found that pioneer species are generally more vulnerable to hurricanes than old-growth specialists (Zimmerman *et al.* 1994). This results is corroborated here by the negative relationship between the severity of hurricane damage and mortality and wood density since early successional species generally have low wood density. What our paper shows is that this relationship may break down under more severe storms, calling for a more nuanced understanding of the association between species successional preferences and the biomechanical characteristics that determine resistance to hurricanes.

We have also added some text in the concluding paragraph to address the need to consider how a more severe hurricane regime will influence composition and successional dynamics of second-growth forests. In past work, we have used models to show that hurricanes of the severity of Hugo will accelerate the mixing of secondary and late successional species by creating opportunities for recruitment (Uriarte *et al.* 2009, Ecological Monographs).

There is a consensus that intense tropical cyclones will likely become more intense because of anthropogenic climate change, and there is some evidence that large cyclones may move more slowly with greater rainfall attending them (e.g., van Oldenborgh *et al.* 2017). Nonetheless, I felt that the authors used terms like “portends” (p2, line 18) and “presages” (p3, line 13) rather incautiously, given the strength of some past Caribbean hurricanes. The authors note the San Felipe hurricane of 1928 (p3, line 13), and Crow (1980) noted that the composition of forests of

the Luquillo Mountains was likely to be shaped not only by that hurricane but also by the San Nicolas and San Ciprian hurricanes of 1931 and 1932. Seen in these terms the effects on intense hurricanes likely happened in the past (see also Figure 1 of Boose *et al.*, 2004, cited cf. Figure 1a of this manuscript). The “extremely active 2017 hurricane season” in the Caribbean is hardly unprecedented either: the likely most intense Caribbean cyclone of the last four centuries (Hurricane San Calixto of 1780) occurred in an active season for hurricanes (all in October that year). It is moot whether “new forests” (per line 6, line 14), affected by a series of intense cyclones, had already developed in response to the cyclone disturbance regime of the 1780s.

Response: We agree with Crow (1980) that the hurricanes of the late 1900s and early 20th century have probably shaped the forests of the Luquillo Mountains but unfortunately we not have the data to assess these potential effects. Nevertheless, even if forests in EYNF had been shaped by these historical storms, the change in the abundance of *P. acuminata* and *C. schreberiana* after H. Hugo suggests that subsequent, more recent storms can have additional selective effects on species composition. On the reviewer’s second point, we do not say that the 2017 season is unprecedented, simply that it was a very active season. It is true that storms of María’s characteristics (or even more severe) have struck the island before but this fact does not invalidate the point that a changing climate is likely to increase the frequency of storms with more severe winds and extreme rainfall.

Minor points:

Terminology: the text alternates between “hurricanes” (a regional term) and “tropical cyclone” (the general meteorological term), sometimes sentence by sentence (e.g., p2, lines 8 vs. 10).

Response: Yes, we use the two terms to alleviate boredom. In the current version, we stick with hurricane unless we are referring to cyclonic storms more generally.

It is untrue that “hurricanes represent the dominant natural disturbance in coastal forests across the Americas” (p2, line 8); cyclone disturbances are very rare in the south Atlantic and most coastal South American forests never experience cyclone disturbance. As for “coastal forests ... in other tropical regions” (p2, line 9), many are unaffected by cyclones (e.g., tropical West Africa).

Response: We have modified the text to address this concern.

I think the title would be more accurate if it sought to generalize less. Because Hurricane María was a single event, the title would be more apt as “A stronger, wetter hurricane tripled stem breaks and doubled tree mortality in a tropical forest” although, even here, the comparative adjectives are problematic. “Stronger and wetter” than what? (see earlier comments about the cyclone disturbance regime).

Response: We have changed the title to address comments from Rev. 1 and 3. It now reads: ‘A stronger, wetter hurricane triples stem breaks and doubles tree mortality in a tropical forest: A comparison of tree damage following three hurricanes.’

Re damage to palms, per earlier comments about using all available data, see Zimmerman & Covich (2007) with respect to Hurricane Georges.

Response: Zimmerman and Covich (2007) only recorded damage data for *P. acuminata* and their plots were elsewhere in the EYNF. As a result, their data is not directly comparable to ours but we cite their work to confirm one of our key findings: the palm is extremely resistant to hurricanes.

Also, re Figures 4, S1, S2, and S4, “PREACU” rather than “PREMON” to be consistent with the main text, and Table S1, *Prestoea acuminata*.

Response: Done.

Tables S1 and S2: Species name is missing for the Guettarda species (GUEVAL) and for the Tetragastris species (TETBAL).

Response: Done. The names and some column headings were erased in the upload process.

For Table S1, www.theplantlist.org lists all of Casearia, Homalium, and Laetia as Salicaceae.

Response: Yes, we were using a previous classification. It is corrected.

Table S2: I presumed the first three columns related to Hurricane Hugo and the next three to Hurricane María, but another row denoting this is needed.

Response: Done. The names and some column headings were erased in the upload process.

I think Table S2 would be improved if it listed values for these species’ wood densities (cf. p4, line 17) and root bole areas (cf. p4, line 21, for the 11 pertinent species).

Response: Done.

P10, reference #15, should be Frangi JL, Lugo AE as authors.

P11, reference #23 should be Lugo AE, Scatena FN as authors.

P12, reference #32: something odd with the author listing.

Response: Done.

References

- Comita LS *et al.*. 2010 Interactive effects of land use history and natural disturbance on seedling dynamics in a subtropical forest. *Ecological Applications* 20, 1270–1284.
- Crow TR 1980 A rainforest chronicle: a 30-year record of change in structure and composition at El Verde, Puerto Rico. *Biotropica* 12, 42–55.
- Uriarte M *et al.*. 2005 Seedling recruitment in a hurricane-driven tropical forest: light limitation, density-dependence and the spatial distribution of parent trees. *Journal of Ecology* 93, 291–304.

van Oldenborgh GJ *et al.*. 2017 Attribution of extreme rainfall from Hurricane Harvey, August 2017. *Environmental Research Letters* 12, 124009
Zimmerman JKH, Covich AP 2007 Damage and recovery of riparian Sierra palms after Hurricane Georges: influence of topography and biotic characteristics. *Biotropica* 39, 43–49.

#####

Reviewer #2 (Remarks to the Author):

This paper presents a novel analysis comparing hurricane damage to trees within a large forest dynamics plot following hurricanes Hugo in 1989 and María in 2017. The analysis shows that stronger, wetter hurricanes such as María do a lot more damage and identifies some species' characteristics that affect vulnerability. Overall, this is an important and timely analysis. There are some relatively minor issues that need to be addressed:

To put the stand-level comparisons in context, it would be helpful to know more about how the pre-Hugo and pre-María stands compared in terms of successional stage/ composition/ history. When was the last major hurricane before Hugo? Was the stand at a similar successional stage for the two hurricanes, with similar size distributions for all species combined, maximum tree sizes, and basal area/stand-level biomass? The analyses in Fig. S2 provide convincing evidence that the size distributions of most of the focal species were similar, and it would be nice to have more context behind the stand-level statistics (e.g., paragraph starting p. 3).

Response: We have added the following text to provide some context for the state of the forest in 1989 when Hugo struck. We now say “The forest has experienced substantial natural and human disturbances during the past century. Prior to 1934 parts of the LFDP were subjected to light logging and agriculture, but the forest structure and canopy cover had substantially recovered when in 1989, after a period of 57 years with no major storm, Hurricane Hugo struck the forest (Thompson *et al.*. 2002). Basal area was estimated to average 36.7 m² ha⁻¹ at the time of hurricane Hugo, 30.85 m² ha⁻¹ at the time Georges struck, and 38.37 m² ha⁻¹ in 2016, the year before María struck the forest.”

We also report maximum tree sizes by species at the time each hurricane struck (Table S4) and size distributions for all species combined (Fig S1). The overall size distributions of trees ≥10 cm dbh at the time of impact did not differ among the three hurricanes (Fig. S1).

Please be more clear in the presentation of mortality rate. It's often confusing whether numbers presented are immediate or total (including delayed) mortality. Furthermore, unless I'm missing something, the paper doesn't actually present the total percent mortality. Please present % immediate and delayed mortality for both storms.

Response: We have clarified how we estimated background, immediate and delayed hurricane-induced mortality. We do not yet have data for delayed mortality from H. María since this process typically plays out over 3-5 years. However, we do provide the requested data (Tables S2-S3& S8).

Similarly, please present total % stem breakage and uprooting.

Response: We now provide this information in Table S2.

This is not a requirement for the paper to be acceptable for publication and may be subject for a separate publication, but it would be interesting to know how damage and mortality varied with tree size (DBH, or height if possible).

Response: The relationship between tree size and damage and mortality figures more prominently in the revised manuscript. We have added values for regression coefficients of size (dbh) for damage and mortality models for both Hugo and María in Table S7 and we now provide a figure (Fig. 4) showing the relationships at the species level in the main manuscript.

Regarding the analysis presented in Figure 4... Presumably, root bole area scales with tree size, so a species mean is likely highly dependent upon the size distribution of trees sampled. Does this match the size distribution within your analysis? Furthermore, to what extent is this relationship driven by root bole area, as opposed to DBH/ height? Would you see a similar relationship if you normalize by DBH (i.e., ratio of root bole area to basal area)?

Response: Our measurements of root bole area were limited and we lack data to assess the effects of the biomechanical properties of trees on damage. Given the concerns expressed by reviewers 2 and 3, we no longer present data on root bole area instead focusing on the effects of tree size on damage.

Minor comments:

-wherever you present results related to mortality (e.g., Fig2a, 3c), specify whether it's current or projected total mortality.

Response: We have made the suggested changes throughout the manuscript and clarified how mortality rates were estimated.

p.2, line 17- you could add 2018...

Response: Yes. Both the 2017 and 2018 seasons were unusually active (with H. Michael, a category 4 storm, in 2018) but we prefer to refer to 2017 for simplicity.

-Fig. S3- why is wind exposure so different? (Please explain what determines the topographic exposure index).

Response: We have added text on the topographic exposure model. We now say 'Exposure to hurricane winds for each tree during each storm was estimated using a topographic model (EXPOS) that determines the degree of exposure to winds given hurricane track and wind speed data and a 5m resolution, LiDAR-derived digital elevation map (DEM). The model assumes that movement over land decreases sustained wind speeds and increases inflow angles, and then calculates spatial variation in exposure at the native resolution of the DEM.'

The main reason for differences in exposure between Hugo and the other two storms is the track of the storms (Fig. 1A). Despite the fact that the maximum wind speed recorded for María was higher than Hugo or Georges, Hurricane María, like Georges, passed to the south side of the Luquillo Mountains, which protected the LFDP forest area on the northern slopes of the mountains from some of the wind force. Hurricane Hugo passed to the North of the Luquillo Mountains so the forest suffered greater exposure to this storm. This text has been added to the legend of Fig. S3.

-p. 8, line 18- from where are background rate estimates obtained?

Response: We have added the following text: “We used data collected in the 1995-1996 census and mixed models to estimate the increase in probability of mortality (i.e., delayed) from background mortality rates that could be attributed to stem breakage or uprooting damage. Random effects for species were included as intercepts (background rates) and as slopes for the damage effect on mortality. Background rates estimate probability of mortality between 1990 and 1995 that cannot be attributed to damage suffered in H. Hugo. Species-specific random slopes for effects of damage on lagged mortality in 1995 estimate increases in probability of mortality that can be attributed to severe damage (i.e., stem break or uprooting) during H. Hugo. Severe damage was coded as a binary variable (0, 1). Note that background mortality rates between 1990 and 1995 are likely overestimates relative to storm-free periods since even if not uprooted or broken stems suffered crown damage that could affect subsequent survival.”

-p. 8, line 16-23- Please be a little more specific/ clear as to how lagged mortality was estimated (e.g., what were the terms in the model, how were model estimates applied to estimate post-María mortality).

Response: See response to previous question.

#####

Reviewer #3 (Remarks to the Author):

This manuscript describes an interesting dataset, comparing damage in the same forest area after two hurricanes with differences in intensity: Hugo, 1989 and María 2017. The premise of the work is that understanding the differences in damage between storms with different wind speeds and rainfall will help us understand how forests will be affected by intensified cyclones that are expected with climate change. I have a number of issues with the way that the data has been collected and is analysed and presented in this manuscript.

General issues

1. Firstly, the title implies that stronger, wetter hurricanes will always triple stem breaks and double tree mortality, while all we have to go on is the difference between two hurricanes. A

more appropriate title might be "A comparison of tree breakage and mortality following two hurricanes".

Response: We now present data for three hurricanes: Hugo, Georges, and María. The new title is: 'A stronger, wetter hurricane triples stem breaks and doubles tree mortality in a tropical forest: A comparison of tree damage and mortality caused by three hurricanes.'

2. The paper uses terminology that is not always clear:

When you say that you assessed mortality in terms of trees uprooted, broken or immediately killed, how did you assess that trees had been killed? Trees can be very badly damaged and still re-sprout the following year. The second survey of mortality six years after Hugo could clarify how accurate the initial assessment was, if the same individual trees were identified in both, but there is nothing to indicate that this was done.

Response: Yes, assessing immediate mortality from hurricanes can be difficult if trees have lost a lot of branches or leaves as a result of wind damage. We characterize mortality as the absence of living tissue remaining anywhere in the individual (i.e., in any sprout or branch). Our approach was very accurate: 98% of the trees assessed as killed in Hugo were dead in the subsequent census of 1995.

3. The term 'tipped up' used on p3 line 17, p7 line 7 and 10, p15 Figure 2 caption is not clear. Do you mean overturned (both broken and uprooted), or just 'uprooted'? If it means 'uprooted' please just use that term consistently.

Response: We now used uprooted throughout the manuscript.

4. How do you define the 'root bole'?

Response: Given reviewer concerns about this aspect of the paper, we have removed all analyses and discussion related to species' root boles.

5. It is not clear what has been measured: In the Methods you say that the trees surveyed following Hugo had stem dbh > 10 inches, while after the María storm surveyed trees were > 10 cm. This is a big difference - Is this an error in the text or an error of the methodology?

Response: This is an error in interpretation. The text said '10cm in dbh'. The 'in' was a preposition but the language was confusing. We have rewritten this sentence.

6. p7, line 10 states that 'we also measured bole size', but doesn't give any details of how this was done. These are complicated structures and would presumably need a consistent measurement system to provide dimensions such as 'bole' spread, root spread, surface area, soil maximum and mean depth, volume etc. What was measured and how?

Response: See response to 4 above.

7. Please include more details of the inputs needed for the topographic model (p7 line 22), and

the scale at which it was applied? Was 'exposure' calculated for each individual tree? Does the exposure value represent exposure through the life of the tree as well as to the damaging hurricane?

Response: See response to reviewer 2. Exposure was calculated for each individual tree. The resolution of the exposure matches that of the topographic layer: 5m.

8. In the Main Text there is a statement that max tree height, leaf area, wood density and root bole area were used in a comparison with mortality / mode of damage. However, the Methods section does not describe how these characteristics were measured. Were they measured on a sample of trees? What sample size? How were they selected? How were they measured? Was it green density you measured?

Response: We provide additional information. We now say 'Wood density (g.cm^{-3}) and specific leaf area (SLA, $\text{cm}^{-2}.\text{g}$) measurements were collected for at least 10 individuals per species using standard protocols³³. Briefly, leaf area was measured on sun-lit foliage of mature individuals and leaves were dried for 48 hours and weighed to calculate SLA. For wood density calculations tree cores were extracted, measured for volume, and oven-dried before weighing. For all analyses, we used the mean trait value for each species.

9. The analysis method does not always appear appropriate. The paper describes a relationship between the change from Hugo to María in proportion of stems broken and the "Species mean root bole area" (Figure 4). However, species bole area means were calculated from trees uprooted (in just one event: María), while the broken trees have no measured root bole area. So you are assuming that the root bole dimensions of uprooted trees tells you something about the root system size of broken trees. As trees break when their breakage moment is reached before their uprooting moment, root or root bole characteristics may be expected to differ between broken and uprooted trees. So the broken trees could have much wider stronger root systems, or perhaps much narrower deeper ones than the uprooted trees.

This will vary between species and soils. Therefore the comparison in Figure 4 does not appear to have validity. To understand the increase in stem breakage in terms of root or root bole characteristics, you would need to have root characteristics from both broken and uprooted trees after both events. There may however be some explanation of the difference in terms of the different distribution of tree sizes between species and events. The BUCCAP and DACEXC species that showed the largest change in proportion of broken stems (Fig 4) appear to have a larger mean increase of stem diameter between events than other species (Fig S2), but unfortunately those differences were not significant.

Response: See response to 4 above.

10. The paragraph (p5) on possible explanations for differences in failure mode appears confused as to whether you expect wet soil to improve or decrease anchorage. Some authors indicate that very wet soils have reduced root-soil cohesion and therefore tree anchorage should be reduced. As the soil under a root plate can shear from the soil under the plate, before roots fail, and the soil weight is just one component of anchorage, it seems unlikely that the increased weight of a wet root plate will make a big enough difference to anchorage to change the failure mode from

uprooting to breakage. Maybe it could, but it would be worth developing an argument by referring to the tree anchorage component work by Coutts from the 1980s. You assume that there really was a difference in soil water content during these two events, but no evidence is presented on this.

Response: We have added an analysis of the relationship between tree size and damage and mortality and how they vary among the two storms. These analyses show that large trees were particularly vulnerable to stem break in María. We now say:

‘Hurricane María combined extreme precipitation and strong winds that may have reduced soil stability and cohesion and thus root anchorage, while simultaneously exerting a strong dynamical force on the stem and crown. Dynamic wind loadings typically drive roots to bend and twist as a result of the rotational pivoting of the trunk, and if the wind is strong enough, it will lead to uprooting²². Pre-hurricane soil moisture in particular is a major controlling factor in the nature of damage (uprooting vs. stem breakage)²³. In dry soils, stem breakage is the dominant type of damage while in wet soils uprooting is more common²³. Given the fact that H. Hugo and Georges were considerable drier than María, it is surprising that uprooting rates were not higher during María than the other two storms. This observation coupled the sharp increase in stems breaks during H. María, particularly for larger trees, may have been driven by the passage of Hurricane Irma on Sept. 7, 2017, two weeks earlier. Irma removed a substantial amount of tree foliage (Zimmerman, pers. obs), possibly reducing wind drag forces over the canopy and how such forces are transmitted to the base of the tree, favoring stem break over uprooting²²’

11. The analysis could be linked with existing work on tree stability and risk. For example, there is no attempt to use or discuss the relevance of published models of root-soil plate/bole dimensions and tree anchorage, or models of wind risk in relation to tree dimensions. Root-bole soil weight increase with higher rainfall could be examined in one of these models to examine the effect on turning moment, to explore your proposal (p5 para 1) that saturated soil in boles led to increased stem breakage. Wind risk models might allow prediction of the increased damage from the observed difference in wind speeds between the hurricanes, and might help understand differences in damage in terms of tree height and other measured characteristics.

Response: As we stated above, we have removed all discussion of the root bole. We lack the data to assess the impacts of this variable and more generally, the biomechanics of trees and their response to wind. Instead, we focus the analyses on the effects of size (dbh) on the nature of wind damage and how these effects differ with storm severity.

Specific points

12. p1 Abstract. References are not usually included in an Abstract. Please remove these.

Response: Nature journals sometimes require referenced paragraphs but Nature Communications does not so we have removed references from the abstract.

13. p4 line 2. Change "in exposure to hurricane-force winds" to "in topographic exposure"?

Response: We have made the requested changes.

14. p4. line 5. Do you mean 'favouring some individuals' rather than 'some species'?

Response: No, we mean species. There are very substantial differences among species. To be more precise, we now say 'by damaging some species more than others'.

15. p4, line 11. strength of the wood

Response: We have made the requested change.

16. p6, line 14-16. Where is the evidence that the reduced species mix would provide reduced C sequestration, wildlife habitat, and other ecosystem services?

Response: We have removed this sentence since we do not examine these factors.

Reviewers' Comments:

Reviewer #1:

Remarks to the Author:

This revised manuscript reports immediate damage and mortality caused by a very intense tropical cyclone to a long-term research site in a tropical forest in Puerto Rico and compares these immediate effects with that of two less intense tropical cyclones 28 and 19 years earlier, in the same forest.

The revised manuscript is improved by including data from after Hurricane Georges, lacking from the earlier version. I appreciated the context provided by analyzing data from periods that lacked severe storms (Table S7).

I liked the new Figure 4 about size relationships with mortality and damage, but felt that this could have been linked more explicitly to traits; a reader needs to cross-reference between this figure and, for example, wood density values in Table S4.

I felt the authors dealt with my point about the secondary forest composition of the study forest rather too cryptically way in the main text (p7, line 22 – the only allusion to it does not really do it). It is well covered in Methods (p8 lines 18–21), although it is hard to reconcile the statement in this manuscript that “parts of the LFDP were subjected to light logging and agriculture” (p8, line 19) with the (still not cited) view of Comita and the current manuscript’s authors (2010) that the northern two-thirds of the LFDP were “subjected to high-intensity historical human land use and, as a result, is dominated by trees of secondary-forest species, such as *Casearia arborea*” (p1272 of that paper). I think a reader needs to know about the history and composition of the Luquillo 16-ha plot explicitly and early (before the Methods section which can then elaborate upon it). This is because the representation of tree traits in the plot, of which the authors make a good deal in this manuscript, may be less than the total species pool could be, biased towards those characteristic of secondary species. This has implications for the generalizability of the study. To link the comment at p7, line 22, I suggest reworking p3 line 10 something like this: “Here we use tree damage and mortality data from secondary tropical forest in Puerto Rico that developed after human disturbance during the first half of the 20th century (Comita et al. 2010) to evaluate the effects of differences... to three storms. Data derive from the 16-ha Luquillo Forest Dynamics Plot (LFDP) after three hurricanes: Hugo in 1989, Georges in 1998, and María in 2017 (Fig. 1A).”

Minor points:

P2, lines 13–15: While these lines have been revised to be more specific, they are still inaccurate. It remains untrue that “cyclonic storms... represent the dominant natural disturbance in coastal tropical forests across the Americas”. Expanding on comments in my earlier review, tropical forests in the Americas on the Atlantic coast from the Orinoco mouth to the Tropic of Capricorn are almost never affected (Hurricane Cara of 2004, category 2, affected coastal Brazil and was a very rare modern exception), and tropical forests on the Pacific coast from about Costa Rica south to Chile are never affected by cyclones, and from south of Mexico to Costa Rica only very rarely.

P5, line 19: delete comma after “wood”

P6, line 11: State what category Hurricane Irma was (it was either 4 or 5 when its eye got to its closest point – 97 km – from Puerto Rico). Since the manuscript doesn’t give a date from Hurricane María in 2017, it seems odd to give one for Irma. The key point is not so much the date but that Irma was just as powerful a hurricane but further from the study site, and exactly two weeks before the more direct hit that was Hurricane María. I think the text could be reworded better to reflect that.

Reference

Comita LS et al. 2010 Interactive effects of land use history and natural disturbance on seedling

dynamics in a subtropical forest. *Ecological Applications* 20, 1270–1284.

Reviewer #2:

Remarks to the Author:

I am satisfied with the responses to my previous comments. I do, however, have several minor comments related to the presentation.

-Be sure to clearly communicate the difference between “wind exposure” and wind speeds experienced. I’d edit the current text on the subject to say something like the following: “Exposure of each tree in the LFDP to wind—AS OPPOSED TO WIND SPEEDS EXPERIENCED—was calculated using the EXPOS model (see methods) for the three hurricanes and exposure of the forest to storm winds was far greater during Hurricane Hugo than during Georges or María (Fig. S3) because of the track of the storms across the island and the position of the study site (Fig. 1A). Consequently, PLOT POSITION RELATIVE TO THE STORM TRACK does not account for the observed differences in damage among storms.”

-Regarding the statement, “The large disparity in the impacts of María compared to the other two storms cannot be explained by differences in tree diameter sizes in the forest at the time of impact (Fig. S1-S2, Table S4).” – Before getting into the results, it may be preferable to make a statement such as, “Despite differences in history, structure and composition of the forest was remarkably similar ahead of each of these hurricanes.”

-On a related note, regarding the paragraph starting with, “The impact of each of the severe hurricanes that have affected the Luquillo forest between 1989 and 2017 may not be independent.” – This may be more than needed in the main text. I don’t have any objections to this paragraph, but it may be possible to reduce length by keeping most of this in the methods (study site description). It also breaks the previous flow of the manuscript, making the following paragraph seem somewhat out of place.

-Regarding the statement, “During H. María however, dense wood, did not afford species any protection from stem break as there was no differences between rates of stem break between species with high and low density wood” – This is an interesting point, which may be worth developing a little more (other than in the last sentence of the manuscript).

-The final paragraph seems to focus too much on uncertainty, as opposed to what we’ve learned from this study.

-Regarding the statement, “Species with higher density wood suffered lower immediate mortality during both Hugo and Maria, were less likely to uproot during María, or break during Hugo (Fig. 3, Table S7).” – awkward/ incorrect grammar.

-Regarding the statement, “Pre-hurricane soil moisture in particular is a major controlling factor in the nature of damage (uprooting vs. stem breakage) 20. In dry soils, stem breakage is the dominant type of damage while in wet soils uprooting is more common 20. Given the fact that H. Hugo and Georges were considerable drier than Maria, it is surprising that uprooting rates were not higher during María than the other two storms.” – Alternatively, is it possible that the prior understanding (from Ref. 20) may be incomplete/ not fully applicable in this setting?

-Fig. S1- Thanks for adding this. It would be more informative with a log y-axis.

Reviewer #3:

Remarks to the Author:

This is important and interesting work comparing damage to forests from hurricanes of differing intensity. The manuscript is much improved from the previous version, and most reviewer suggestions have been addressed. I have a 'major' comment on the structure of this version, and a few very 'minor' suggested edits below:

Major comment

The structure of the manuscript is very difficult to follow, with Introduction, Results and Discussion all mixed together. Please revise the structure to follow the Nature Communications "Article" structure as indicated at www.nature.com/ncomms/submit/article. This says that the Main text should include separate Introduction, Results, Discussion (if appropriate), and Methods (if appropriate) sections. I suggest that a separate Discussion section would be helpful. Please also include a hypothesis statement in the Introduction.

Minor edits

p2 line 21. Please change Km to km.

p3 line 13. I suggest you delete "risk".

p7 line 17. I suggest you change "Forecasting" to "Projecting". The term "forecast" is not normally used to describe future climates.

p9 line 5. Please change "10 in diameter" to "10 cm diameter".

p9 line 11. Add space before cm.

p10 line 6. Add space before m.

REVIEWERS' COMMENTS

Reviewer #1 (Remarks to the Author)

This revised manuscript reports immediate damage and mortality caused by a very intense tropical cyclone to a long-term research site in a tropical forest in Puerto Rico and compares these immediate effects with that of two less intense tropical cyclones 28 and 19 years earlier, in the same forest.

The revised manuscript is improved by including data from after Hurricane Georges, lacking from the earlier version. I appreciated the context provided by analyzing data from periods that lacked severe storms (Table S7).

I liked the new Figure 4 about size relationships with mortality and damage, but felt that this could have been linked more explicitly to traits; a reader needs to cross-reference between this Figure and, for example, wood density values in Table S4.

Response: We tried using a graded color scale for wood density in a previous version of this figure but the resulting figure was very hard to see. We have increased the width of the species' lines to facilitate visualization.

I felt the authors dealt with my point about the secondary forest composition of the study forest rather too cryptically way in the main text (p7, line 22 – the only allusion to it does not really do it). It is well covered in Methods (p8 lines 18–21), although it is hard to reconcile the statement in this manuscript that “parts of the LFDP were subjected to light logging and agriculture” (p8, line 19) with the (still not cited) view of Comita and the current manuscript's authors (2010) that the northern two-thirds of the LFDP were “subjected to high-intensity historical human land use and, as a result, is dominated by trees of secondary-forest species, such as *Casearia arborea*” (p1272 of that paper). I think a reader needs to know about the history and composition of the Luquillo 16-ha plot explicitly and early (before the Methods section which can then elaborate upon it). This is because the representation of tree traits in the plot, of which the authors make a good deal in this manuscript, may be less than the total species pool could be, biased towards those characteristic of secondary species. This has implications for the generalizability of the study.

To link the comment at p7, line 22, I suggest reworking p3 line 10 something like this: “Here we use tree damage and mortality data from secondary tropical forest in Puerto Rico that developed after human disturbance during the first half of the 20th century (Comita et al. 2010) to evaluate the effects of differences... to three storms. Data derive from the 16-ha Luquillo Forest Dynamics Plot (LFDP) after three hurricanes: Hugo in 1989, Georges in 1998, and María in 2017 (Fig. 1A).”

Response: We have made the requested change although we cite Thompson et al (2002) rather than Comita et al. (2010). The former is a better reference for the land use history of the study site.

Minor points:

P2, lines 13–15: While these lines have been revised to be more specific, they are still inaccurate. It remains untrue that “cyclonic storms... represent the dominant natural disturbance in coastal tropical forests across the Americas”. Expanding on comments in my earlier review, tropical forests in the Americas on the Atlantic coast from the Orinoco mouth to the Tropic of Capricorn are almost never affected (Hurricane Cara of 2004, category 2, affected coastal Brazil and was a very rare modern exception), and tropical forests on the Pacific coast from about Costa Rica south to Chile are never affected by cyclones, and from south of Mexico to Costa Rica only very rarely.

Response: We now say ‘Cyclonic storms (hurricanes, cyclones, and typhoons) represent the dominant natural disturbance in coastal tropical forests across the Caribbean, the Indian subcontinent, Southeast Asia, Indo-Malaysia, and northern Australia’.

P5, line 19: delete comma after “wood”

Response: Done.

P6, line 11: State what category Hurricane Irma was (it was either 4 or 5 when its eye got to its closest point – 97 km – from Puerto Rico). Since the manuscript doesn’t give a date from Hurricane María in 2017, it seems odd to give one for Irma. The key point is not so much the date but that Irma was just as powerful a hurricane but further from the study site, and exactly two weeks before the more direct hit that was Hurricane María. I think the text could be reworded better to reflect that.

Response: We now say ‘This observation coupled the sharp increase in stem breaks during H. María, particularly for larger trees (Fig. 4), may have been driven by the passage of Hurricane Irma, a category 4 storm that skirted the north of the island on Sept. 7, 2017, two weeks before Sept. 20, when María struck the island. Although Irma did not make landfall on Puerto Rico, it removed a substantial amount of tree foliage (Zimmerman, pers. obs), possibly reducing wind drag forces over the canopy and how such forces are transmitted to the base of the tree, favoring stem break over uprooting²⁰.’

Reference

Comita LS et al. 2010 Interactive effects of land use history and natural disturbance on seedling dynamics in a subtropical forest. *Ecological Applications* 20, 1270–1284.

Reviewer #2 (Remarks to the Author)

I am satisfied with the responses to my previous comments. I do, however, have several minor comments related to the presentation.

-Be sure to clearly communicate the difference between “wind exposure” and wind speeds

experienced. I'd edit the current text on the subject to say something like the following:
"Exposure of each tree in the LFDP to wind—AS OPPOSED TO WIND SPEEDS EXPERIENCED—was calculated using the EXPOS model (see methods) for the three hurricanes and exposure of the forest to storm winds was far greater during Hurricane Hugo than during Georges or María (Fig. S3) because of the track of the storms across the island and the position of the study site (Fig. 1A). Consequently, PLOT POSITION RELATIVE TO THE STORM TRACK does not account for the observed differences in damage among storms."

Response: Done.

-Regarding the statement, "The large disparity in the impacts of María compared to the other two storms cannot be explained by differences in tree diameter sizes in the forest at the time of impact (Fig. S1-S2, Table S4)." – Before getting into the results, it may be preferable to make a statement such as, "Despite differences in history, structure and composition of the forest was remarkably similar ahead of each of these hurricanes."

Response: Done.

-On a related note, regarding the paragraph starting with, "The impact of each of the severe hurricanes that have affected the Luquillo forest between 1989 and 2017 may not be independent." – This may be more than needed in the main text. I don't have any objections to this paragraph, but it may be possible to reduce length by keeping most of this in the methods (study site description). It also breaks the previous flow of the manuscript, making the following paragraph seem somewhat out of place.

Response: We have moved this text to the discussion.

-Regarding the statement, "During H. María however, dense wood, did not afford species any protection from stem break as there was no differences between rates of stem break between species with high and low density wood" – This is an interesting point, which may be worth developing a little more (other than in the last sentence of the manuscript).

Response: We have added some text to the discussion.

-The final paragraph seems to focus too much on uncertainty, as opposed to what we've learned from this study.

Response: There is tremendous uncertainty about the impacts of more severe storms on these forests so the tone of the paragraph reflects where we stand. Nevertheless, we have added some text in the discussion section to highlight our findings.

-Regarding the statement, "Species with higher density wood suffered lower immediate mortality during both Hugo and Maria, were less likely to uproot during María, or break during Hugo (Fig. 3, Table S7)." – awkward/ incorrect grammar.

Response: We now say ‘Species with high wood density suffered lower immediate mortality during both Hugo and Maria and were less likely to uproot during María, or break during Hugo (Fig. 3, Table S7).’

-Regarding the statement, “Pre-hurricane soil moisture in particular is a major controlling factor in the nature of damage (uprooting vs. stem breakage). In dry soils, stem breakage is the dominant type of damage while in wet soils uprooting is more common. Given the fact that H. Hugo and Georges were considerable drier than Maria, it is surprising that uprooting rates were not higher during María than the other two storms.” – Alternatively, is it possible that the prior understanding (from Ref. 20) may be incomplete/ not fully applicable in this setting?

Response: Yes, as we explain in the text, we believe that leaf removal during Irma may underlie this observation. Nevertheless, we have added some text to account for the possibility that other unaccounted factors may underlie this observation.

-Fig. S1- Thanks for adding this. It would be more informative with a log y-axis.

Response: We have made the requested change.

#####

Reviewer #3 (Remarks to the Author)

This is important and interesting work comparing damage to forests from hurricanes of differing intensity. The manuscript is much improved from the previous version, and most reviewer suggestions have been addressed. I have a 'major' comment on the structure of this version, and a few very 'minor' suggested edits below:

Major comment

The structure of the manuscript is very difficult to follow, with Introduction, Results and Discussion all mixed together. Please revise the structure to follow the Nature Communications "Article" structure as indicated at www.nature.com/ncomms/submit/article. This says that the Main text should include separate Introduction, Results, Discussion (if appropriate), and Methods (if appropriate) sections. I suggest that a separate Discussion section would be helpful. Please also include a hypothesis statement in the Introduction.

Response: We have changed the structure as suggested by the referee and editor. We now have an introduction, results with sub-headings, and a discussion section. We have not included a hypothesis statement because doing so would be a *post hoc* exercise with little meaning, given the lack of previous tree damage and mortality data for a storm of this severity and the context dependence of impacts.

Minor edits

p2 line 21. Please change Km to km.

Response: Done.

p3 line 13. I suggest you delete "risk".

Response: We prefer to keep it as it reflects what we aim to achieve.

p7 line 17. I suggest you change "Forecasting" to "Projecting". The term "forecast" is not normally used to describe future climates.

Response: Done.

p9 line 5. Please change "10 in diameter" to "10 cm diameter".

Response: Done.

p9 line 11. Add space before cm.

Response: Done.

p10 line 6. Add space before m.

Response: Done.